# Performance Evaluation of Hospital Site Suitability Using Multilayer Perceptron (MLP) and Analytical Hierarchy Process (AHP) Models in Malacca, Malaysia

Khaled Yousef Almansi [1,*], Abdul Rashid Mohamed Shariff [2] , Bahareh Kalantar [3] , Ahmad Fikri Abdullah [2], Sharifah Norkhadijah Syed Ismail [4] and Naonori Ueda [3]

1   Department of Civil Engineering, Faculty of Engineering, Universiti Putra Malaysia (UPM), Serdang 43400, Malaysia
2   Department of Biological and Agricultural Engineering, Faculty of Engineering, Universiti Putra Malaysia (UPM), Serdang 43400, Malaysia; rashidpls@upm.edu.my (A.R.M.S.); ahmadfikri@upm.edu.my (A.F.A.)
3   RIKEN Center of Advanced Intelligence Project, The Goal-Oriented Technology Research Group, Disaster Resilience Science Team, Tokyo 103-0027, Japan; bahareh.kalantar@riken.jp (B.K.); naonori.ueda@riken.jp (N.U.)
4   Department of Environmental and Occupational Health, Faculty of Medicine and Health Sciences, Universiti Putra Malaysia (UPM), Serdang 43400, Malaysia; norkhadijah@upm.edu.my
*   Correspondence: gs47515@student.upm.edu.my

**Abstract:** This study focuses on suitable site identification for constructing a hospital in Malacca, Malaysia. Using significant environmental, topographic, and geodemographic factors, the study evaluated and compared machine learning (ML) and multicriteria decision analysis (MCDA) for hospital site suitability mapping to discover the highest influential factors that minimize the error ratio and maximize the effectiveness of the suitability investigation. Identification of the most significant conditioning parameters that impact the choice of an appropriate hospital site was accomplished using correlation-based feature selection (CFS) with a search algorithm (greedy stepwise). To model the potential hospital site map, we utilized multilayer perceptron (MLP) and analytical hierarchy process (AHP) models. The outcome of the predicted site models was validated utilizing CFS 10-fold cross-validation, as well as ROC curve (receiver operating characteristic curve). The analysis of CFS indicated a very high correlation with R2 values of 0.99 for the MLP model. However, the ROC curve indicated a prediction accuracy of 80% for the MLP model and 83% for the AHP model. The findings revealed that the MLP model is reliable and consistent with the AHP. It is a sufficiently promising approach to the location suitability of hospitals to ensure effective planning and performance of healthcare delivery.

**Keywords:** GIS; hospital site suitability; multilayer perception (MLP); analytical hierarchy process (AHP)

## 1. Introduction

Hospitals have become the most critical facilities where people go for health-related issues and to improve their health standards. Over the past three years, the need for hospitals has tremendously increased due to the COVID-19 pandemic, which has revealed enormous shortcomings in hospitals worldwide [1]. Many countries and institutions give great focus to hospital site selection. Selecting a suitable hospital site in any area with some specific selected factors is a very challenging procedure. It is not just the technical build-up, but also social, environmental, and contradictory political points. The location of a new service, such as a hospital location, is a significant decision-making problem for both urban planners and decision makers. Choosing the most suitable location from several alternative

sites is demanding and complex. Site selection is a decision-making process that requires the weighing of criteria and the evaluation and classification of alternatives [2,3].

Over time, the need for healthcare facilities has spontaneously increased across the globe [4,5]. Health care services have a vital role in the socio-economic growth of any country. The principal purpose of medical services is to accomplish the demand for health care establishments for everyone at any given time [6–8]. The necessity for medical services is increasing in areas of urban expansion due to constant migration from rural areas, which leads to a difference in the admission of these services [9–11].

It is necessary to establish more hospitals to enhance healthcare facilities. The first step to accomplishing this is to choose the sites to establish the hospitals. Choosing hospital locations is a many-sided issue. It involves considerable interveners, such as patients, doctors, healthcare staff, and real estate developers, and needs suitable harmonization among authorities, urban planners, and health-affiliated policymakers [12–14]. Optimal location of a hospital will enhance the operation of hospitals in terms of service delivery. Unsolicited location selection consistently results in expanded costs and a reduction in customer fulfillment. Accordingly, to fulfill improved healthcare facility requirements, it is necessary to appoint new optimal hospital locations.

Choosing optimal hospital locations hinges on several factors. These factors are heterogeneous and involve an optimization technique to assess their influences [15,16]. A multidimensional method for making a decision is needed that can be satisfied using the machine learning (ML) method [17,18]. A geographic information system (GIS) integrated with ML can help in site selection by analyzing its spatial dimensions. This capability allows urban pioneers to make a refined decision regarding the selection of hospital sites. This combination would be well-suited when planning mechanisms do not provide prescriptive procedures for the findings. A recent study was conducted to choose optimal hospital locations using ML. The study was carried out by applying different ML classifications, such as support vector machine (SVM), multilayer perceptron (MLP), and linear regression (LR) [19]. Given the above discussion, establishing new hospitals is undeniably a basic requirement. Hospital site selection is challenging and needs to be supported by cutting edge decision-making techniques, such as ML and GIS. A gap exists in the literature on this topic, so a detailed assessment of hospital site selection parameters and criteria including ML classifications is necessary. For this study, an MLP model was selected based on the fact that it was successfully applied in hospital site suitability assessment and proven to be the most effective model. This study validates the effectiveness of the MLP model in a new study area and compares it with the analytical hierarchy process (AHP), the most widely applied model in the Multiple-Criteria Decision Analysis (MCDA) literature.

This study aims to evaluate and compare MLP and AHP as evidence of the significant influence of ML methods in hospital site suitability. For this purpose, in the first stage, three main criteria and 14 sub-criteria are selected according to the literature review and study area characteristics. The subsequent stage includes the geographical mapping of parameters utilizing a GIS environment. The next stage involves the GIS-based multicriteria decision analysis (MCDA) and ML algorithms, which comprise prioritizing the main criteria and sub-criteria utilizing MLP and AHP. In the penultimate stage, two-site suitability maps are produced to govern the ideal locations for the hospital. Finally, a critical comparison is made between the MLP and AHP methods for the hospital site selection process. The objectives of this study are:

1.  To investigate suitable sites for establishing new hospitals in Malacca;
2.  To identify and map relevant environmental, topographic, and geodemographic conditioning factors and discover their weighted commitment in selecting suitable sites for new hospitals;
3.  To identify the most-influencing factors that impact the choice of a suitable hospital location using correlation-based feature selection (CFS) and a search algorithm (greedy stepwise);

4.  To apply MLP, AHP, and weighted overlay analysis to prepare hospital site suitability maps;
5.  To validate the results of the suitability maps based on sensitivity, specificity, area under the curve (AUC), and 10-fold cross-validation.

## 2. Literature Review

### 2.1. MCDA Technique for Site Selection

Several studies have used MCDA methods to analyze the complexity involved in assessing site suitability. Hopkins (1977) provided a comparative evaluation for site suitability methods as a methodological basis in suitability analyses. The study indicated several key points: (1) A spatial suitability model should be used with an emphasis on cartographic modeling to identify areas suitable for a particular use and to determine the suitability of an area for a particular use. (2) To determine the suitability of a particular area, the suitability modeling must include several criteria. (3) When multiple criteria and conflicting priorities appear, it is necessary to use a multi-criteria analysis [3].

MCDA offers a range of measures and procedures to unravel challenging decision-making issues in hierarchy order [20]. AHP is among the most-used MCDA methods for site selection [21,22]. It uses deterministic data values to analyze decision making. However, the data are often incomplete and intricate. Experts create different opinions from different perceptions. This initiates doubt and hesitation in making a decision [23]. The MCDA method is regularly employed in the assessment of location selected because it is a multidimensional solution that contains various factors in decision making. The issues of evaluating the location of a facility are among the critical problems in decision making. It is a decisive mission of the authorities to solve land tenure problems, avert unwanted environmental burdens, and exploit the profitability of land use [24–26]. Site selection supported by GIS could have an essential function in creating an environment to solve spatial data problems [27]. Though GIS and MCDA are two diverse areas of study, their incorporation could profit the complexity of location selection by investigating the geographical decision and assessing the order of other factors [8,28]. Site selection studies, such as railway stations [29,30], fire stations [31,32], solar photovoltaic plants [33,34], regional landfills [35,36], and disposal of wastewater [37], have validated that MCDA could be the most suitable model for solving site-based issues (Table 1).

Choosing a hospital site is one of the difficulties that planners face, particularly in developed countries [38,39]. Several research articles concentrated on hospital site selection have been discovered. The literature has presented several methods used for the selection of hospital sites. For instance, [7,40–42] have projected a GIS-based technique for the selection of hospital sites. However, they have not taken into account the importance of the standards influencing hospital location and concentrated solely on GIS analysis. Consequently, an MCDA GIS-based method was conducted to select hospital sites that contained the priorities of influenced factors and their spatial allocation. This method can achieve evident scientific decision making for the selection of hospital locations. Various factors can be considered which cannot be ignored for a realistic solution. These factors differ by site [43,44]. Critical parameters that influence hospital locations are specified through the literature review. These parameters are altitude, aspect, slope, curvature, topographic roughness index (*TRI*), topographic wetness index (*TWI*), stream power index (*SPI*), proximity to roads, proximity to highways, distance to river networks, distance to residential areas, distance to agricultural areas, population size, and population density [45–49].

Additionally, when choosing influence factors, it is important to define their relationships to one other and their levels of influence on the overall site suitability.

**Table 1.** List of MCDA models and main criteria used in the literature.

| Name | MCDA Model | Decision Problem and Criteria Used |
| --- | --- | --- |
| Vahidnia et al. [50] | Fuzzy AHP | Prioritizing hospital location for target population with minimum time, pollution, and cost. |
| Alavi et al. [51] | AHP & TOPSIS | Determining optimal location of hospitals based on road access factors and green spaces, as well as distance from industrial and military centers. |
| Abdullahi et al. [52] | AHP & OLS | Comparing AHP and the ordinary least square (OLS) evaluation model based on technical, environmental, and socio-economic factors for selecting new suitable sites. |
| Ahmed et al. [12] | AHP | Determining the optimal location of a new hospital based on urban, environmental, and economic factors. |
| Rahimi et al. [53] | AHP | Determining optimal locations for hospitals based on urban land and social factors. |
| Youzi et al. [49] | AHP | Determining the optimal location of a new hospital based on the criteria of utility, performance, safety, population, density, proximity, and adaptability factors. |
| Soltani et al. [26] | AHP | Choosing optimal sites for hospitals based on spatial analysis and urban land use planning factors. |
| Kahraman et al. [54] | Fuzzy TOPSIS | Developing spherical fuzzy TOPSIS and applying it to a hospital site selection problem. |
| Tripathi et al. [55] | AHP & Fuzzy AHP | Determining a suitable MCDA method for selecting hospital sites on a social, geographic, and environmental basis. |

### 2.2. ML for Site Suitability

ML has been widely used in the field of geohazards, and several works have explored the capability of ML for spatial data analysis for landslides [56,57], land suitability, flood hazards [58], water applications, wind, and energy [48,59–63].

Few studies have applied ML models to assess site suitability (Table 2). Three ML models (MLP, SVM, and LR) were introduced to hospital site suitability. The proposed methodology was tested in Palestine, and the results showed that the MLP model achieved a higher performance, which shows that the model is suitable and promising for assessing the suitability of hospital sites [19]. Yang [64] developed a web-based GIS platform and ML technique package called HOLSAT (Hotel Location Selection and Analyzing Toolset) that can be utilized to predict suitable sites for building hotels, and the system has been successfully used in the tourism industry for locating optimal spaces for hotel sites. Another study developed a framework based on a multilayer feed-forward neural network to generate suitability maps at a regional scale for the identification of potentially safe landfill sites. The results reliably identified suitable and safe locations for effective waste management [65]. In a related work [66], the authors developed an ArcGIS spatial data mining toolbox based on MLP neural networks to detect suitable sites for landfills. The toolbox was tested using a dataset from the northern states of Malaysia, and the results showed that the toolbox simplified the process of landfill site selection across the states. A study also proposed MLP-BP and fuzzy inference methodology to optimally rank locations of large hydro power plant according to level of suitability. Overall, the results produced a comprehensive classification of location suitability and the best alternatives among the sites [67].

**Table 2.** List of ML models of site suitability used in the literature.

| Name | ML Model | Decision Problem |
|---|---|---|
| Yang et al. [64] | MLP, LR, SVR, PPR, & BR | Hotel location selection |
| Abujayyab et al. [65] Abujayyab et al. [66] | MLP and ANN | Landfill site suitability |
| Shimray et al. [67] | MLP-BP | Hydro power plant site selection |
| Al-Ruzouq et al. [68] | FR, BTR, SVM, & AHP | Dam site suitability |
| Taghizadeh-Mehrjardi et al. [69] | SVM & RF | Land suitability and the sustainability of agricultural production |
| Almansi et al. [19] | MLP, SVM, & LR | Hospital site suitability |

## 3. Materials and Methods

### 3.1. Study Area

Malacca is a state in the central region of Malaysia (Figure 1); the state has an approximate area of 1664 square kilometers and a population of 934,600 people [70]. Malacca is divided into three regions (Jasin, Alor Gajah, and Malacca Tengah),and is the smallest administrative unit in any state in Malaysia. Malacca Tengah has taken the lead towards achieving the status of city network state and main development corridor in Malacca. Furthermore, Malacca Tengah is the economic hub, with tourism, education, administration, and employment opportunities [71]. Geographically, Malacca has a flat topography and is drained by rivers flowing northeast–southwest through the Strait of Malacca. There are areas in the northern region with high elevation, such as Bukit Manis (169 m), Hill End (210 m), Bukit Punggur (397 m), and Bukit Batang Malacca (433 m). Terrain analysis shows that about 0.10% of the land lies on a critical slope > 25 degrees, and those areas are unsuitable for development. Agriculture is the most-widespread land use in Malacca, as well as settlement, industrial, institutional, and business [71].

### 3.2. Data Description and GIS Techniques

3.2.1. Hospital Sites

Prediction tasks using data mining require a process of using existing data to classify new datasets by determining the relationship (weight) between the input data (variables). In this study, hospital locations comprised the dependent variable, i.e., the tested and measured variable, upon which the weight and relationship of the independent variables (conditioning factors) were discovered. Eight hospital locations were collected from Jabatan Kesihatan Negeri Melaka for Malacca (Figure 2). In Melaka, there are three government hospitals (one for each district) and five private hospitals, all located in Melaka Tengah.

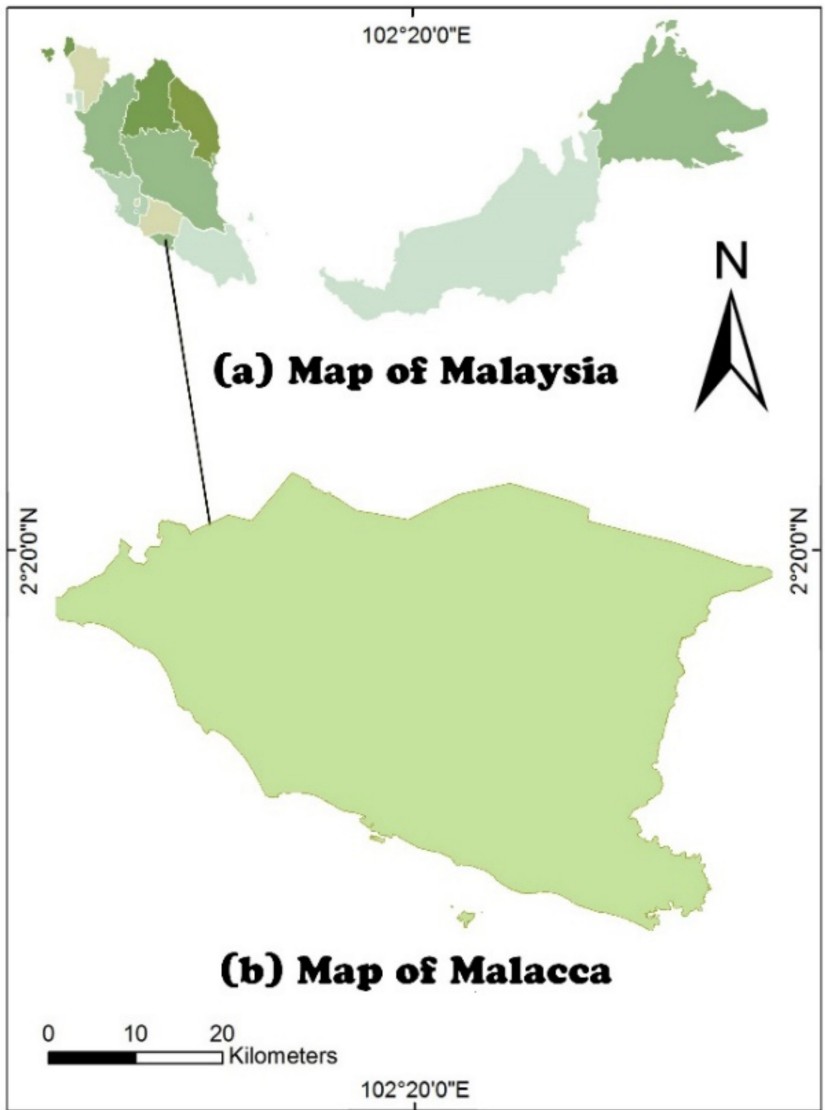

**Figure 1.** (**a**) Map of Malaysia and (**b**) map of Malacca.

3.2.2. Conditioning Factors

To assess the suitable sites for hospital locations in Malacca, datasets obtained in numerical format from different institutions were compiled in a GIS environment. The conditioning parameters utilized were classified into three categories: environmental, topographic, and geodemographic factors. According to the literature, researchers have no agreement on precise parameters for location suitability assessments. Nevertheless, some parameters that researchers have used extensively indicate their preference in location-based decision making [72]. From the classifications, layers of data related to the study area were prepared and further processed. The conditioning parameters were converted to raster resolution of 10 m using ArcGIS 10.5.

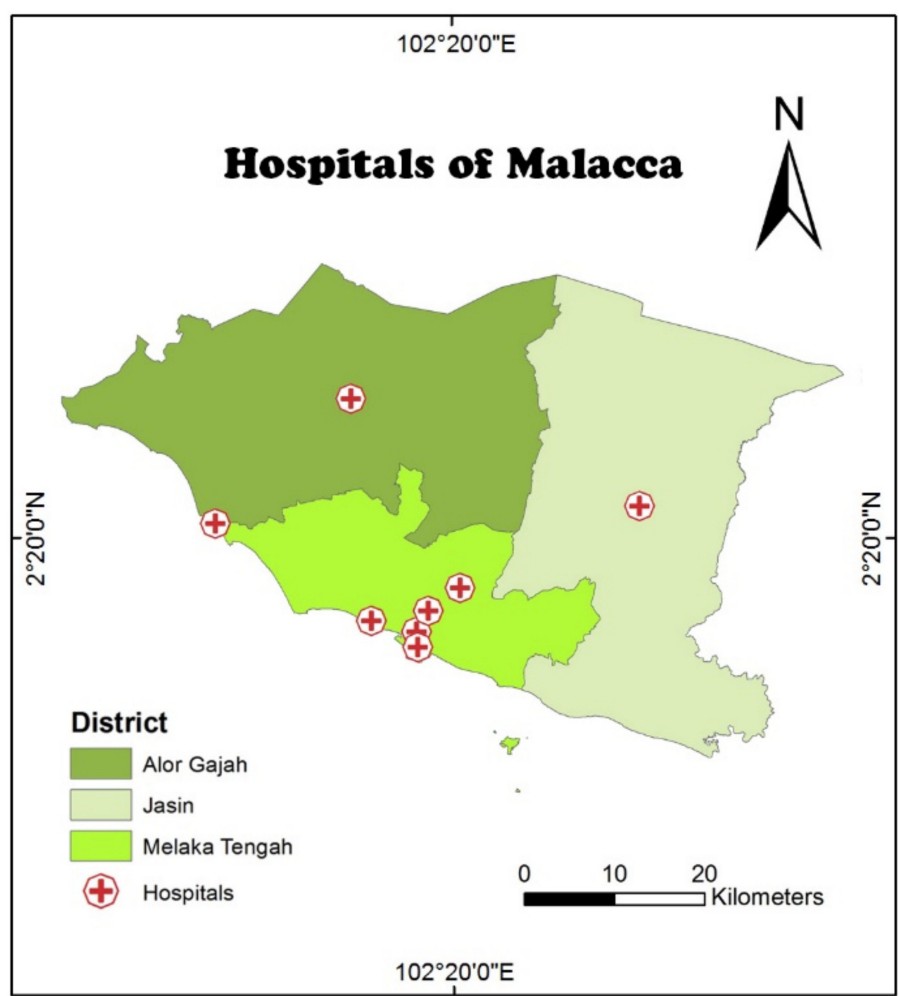

**Figure 2.** Hospitals of Malacca.

The terrain of an area directly influences location selection because it handles many of the natural processes that shape environmental processes, such as floods and erosion. In this study, seven topographical factors applicable to evaluating the suitability of hospital sites were identified: aspect, altitude, slope, curvature, *SPI*, TRI, and *TWI* (Figure 3a–g).

Human settlement, such as landscape, availability of water, vegetation and woodland distribution, and productive soil for the production of crops, is primarily controlled by natural phenomena. For environmental variables, five data layers were derived: roads, highways, river networks, residential areas, and agricultural areas of land use or land cover. Euclidean distance was used to extract the distance from residential areas, agricultural areas, river networks, roads, and highways (Figure 3h–l). Euclidean distance quantifies the relation between the proper site and the conditioning factors in linear distance [46,48,73,74].

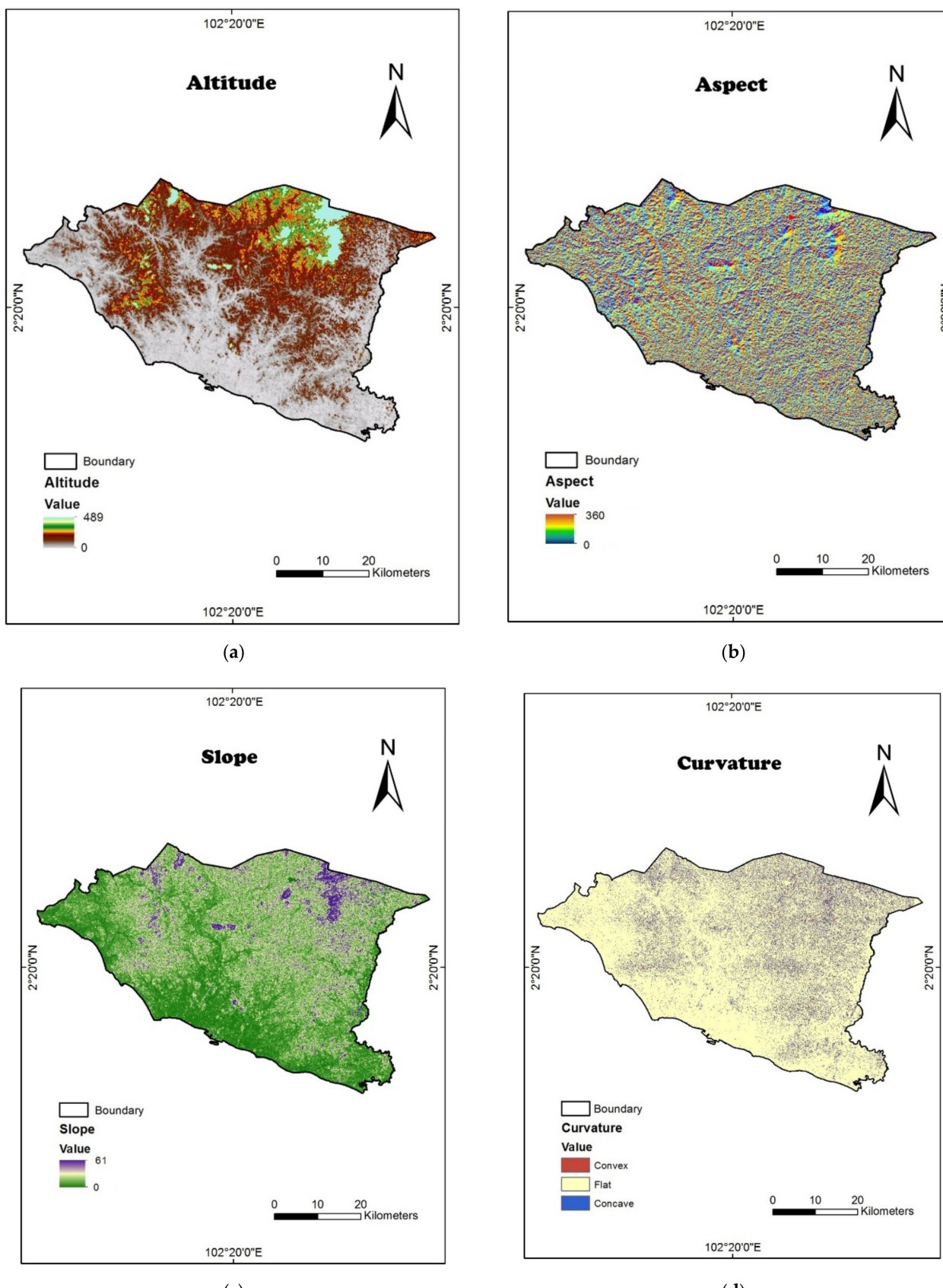

**Figure 3.** *Cont*.

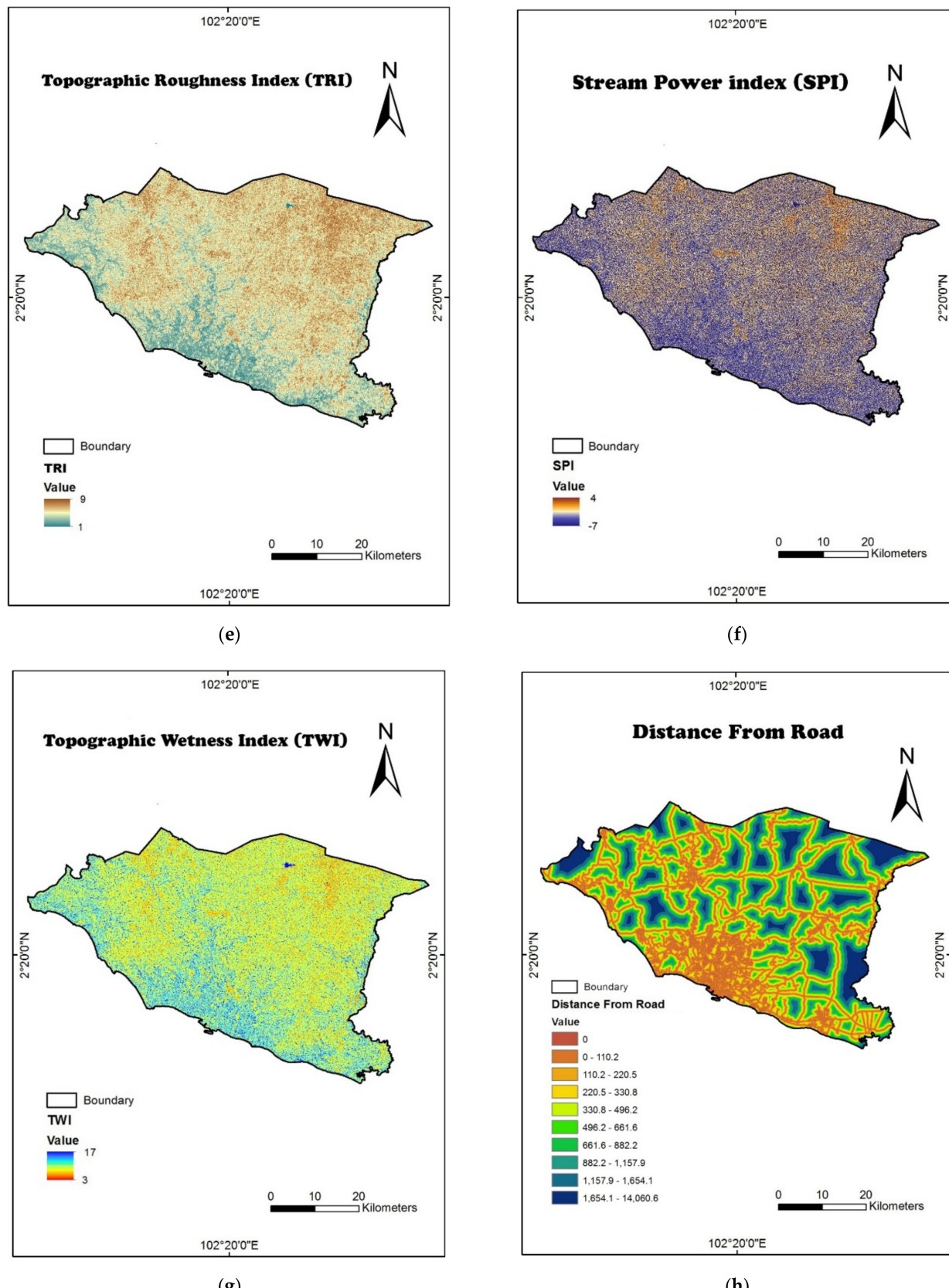

**Figure 3.** *Cont*.

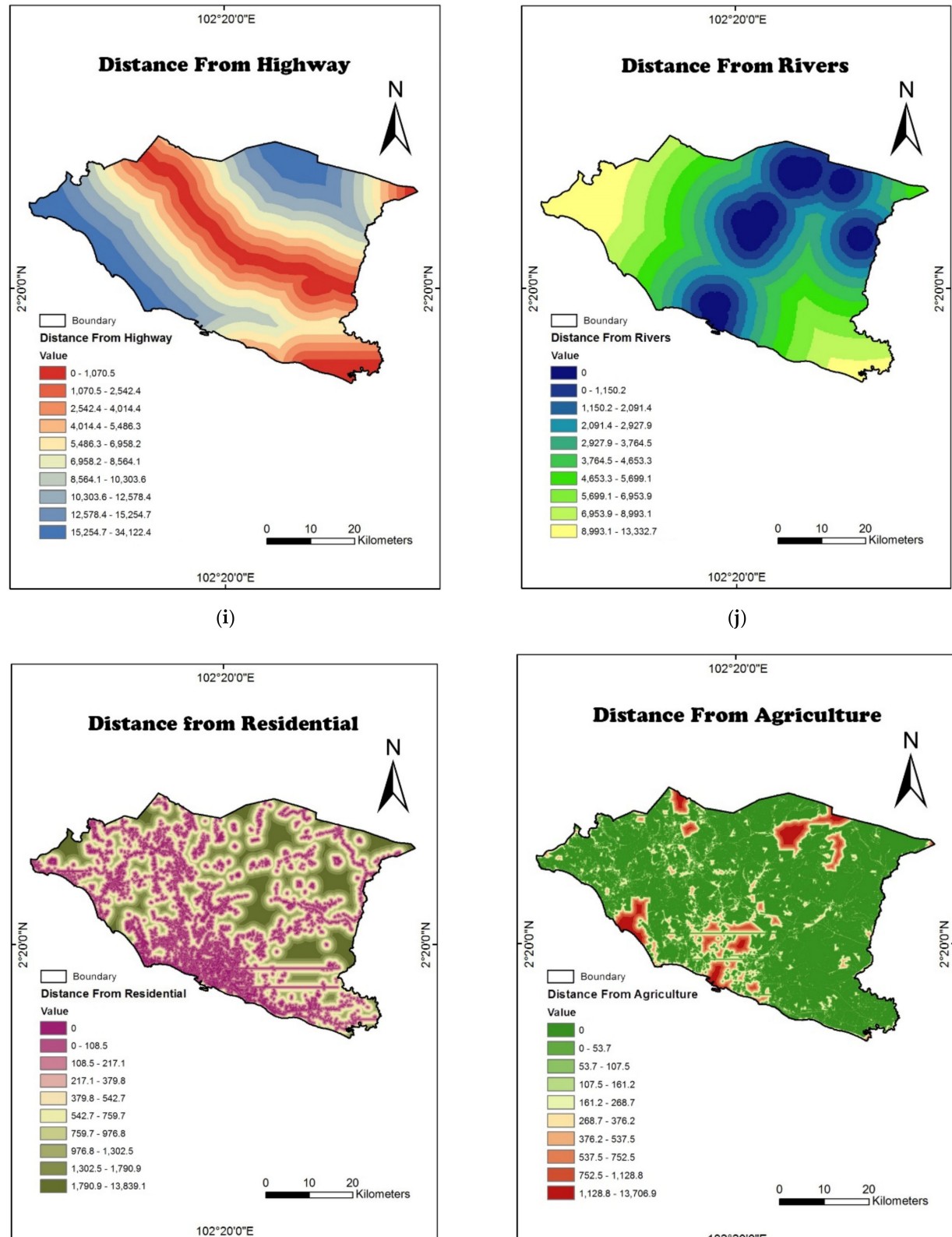

**Figure 3.** *Cont.*

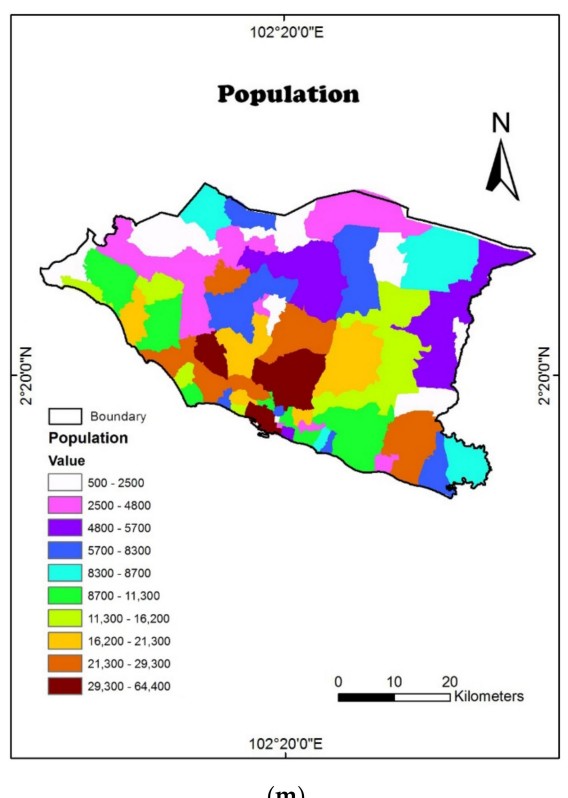

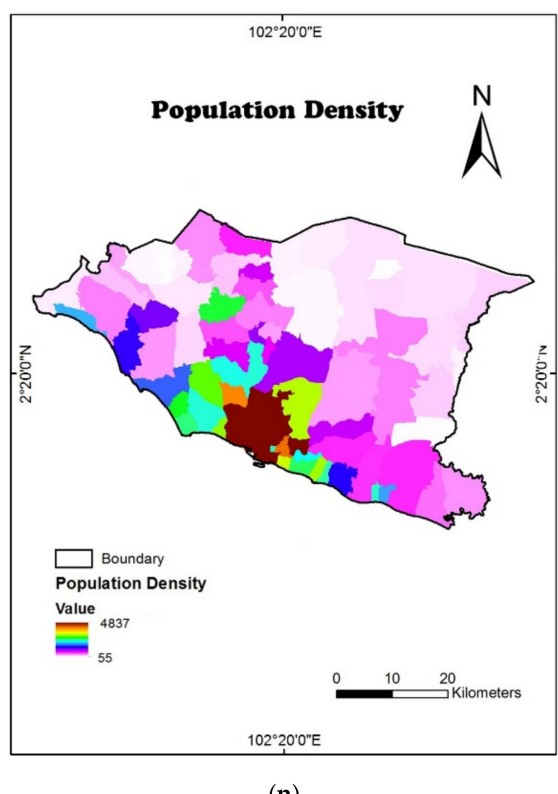

(**m**)                                                                 (**n**)

**Figure 3.** Malacca conditioning factors: (**a**) altitude, (**b**) aspect, (**c**) slope, (**d**) curvature, (**e**) distance from road, (**f**) distance from highway, (**g**) distance from river, (**h**) distance from residential, (**i**) distance from agriculture, (**j**) *TRI*, (**k**) *SPI*, (**l**) *TWI*, (**m**) population, and (**n**) population density.

Hospital facilities close to where people live directly affect emergency response during disasters in terms of cost and response time. In any society, the population is not populated evenly, and this explains the variance in population density, along with various societal measurements that impact the selection of locating a new hospital [49,75–77]. The combination of demographic and geographic parameters qualifies the inclusion of population size and population density as significant parameters to deliver an extensive basis for studying and developing health care services. In this study, the Malacca case study utilized 30 m resolution ASTER GDEM data acquired in 2015 and obtained from Tindak Malaysia. Other relevant data, such as Mukim boundaries, district boundaries, highways, road networks, rivers, residential areas, and agricultural areas were obtained from Jabatan Perancangan Bandar Dan Desa (Figure 3m,n).

*3.3. Hospital Site Suitability Conditioning Factors*

Conditioning factors is the general name used to describe a range of parameters (data, value, or conditions) that favor selecting space as suitable to locate facilities [46]. The correlations between conditioning factors and hospital sites were examined in this study for suitability analysis. To achieve this, a spatial database containing the conditioning factors selected from the literature sources (specifically, from the works of [46,48,72,78]) was prepared and constructed. First, all the identified influential conditioning factors considered in our study were listed (Table 3) and analyzed as data layers for the study area of Malacca to produce the independent variables.

**Table 3.** List of the conditioning factors used.

| No. | Conditioning Factors |
|---|---|
| 1 | Altitude |
| 2 | Slope |
| 3 | Aspect |
| 4 | Curvature |
| 5 | *SPI* |
| 6 | *TWI* |
| 7 | *TRI* |
| 8 | Distance from river network |
| 9 | Distance from highway |
| 10 | Distance road |
| 11 | Distance from the agricultural area |
| 12 | Distance from the residential area |
| 13 | Population size |
| 14 | Population density |

The dataset parameters exist in continuous formats, e.g., elevation, slope, TRI, and categorized data (such as land use). However, for successful analysis, it was necessary to convert the parameters to an organized data format. Therefore, the conditioning factors were converted to categorize data classified into 10 classes (adopted from the maximum number of classes in the literature). An exception was curvature, which was classified into three classes: concave, convex, and flat (Table 4) [46].

**Table 4.** Conditioning factor suitability rating.

| Factors | Classes | Suitability Rating |
|---|---|---|
| Altitude (m) | 0–9 | 1 |
| | 9–12 | 2 |
| | 12–16 | 3 |
| | 16–21 | 4 |
| | 21–27 | 5 |
| | 27–34 | 6 |
| | 34–43 | 7 |
| | 43–56 | 8 |
| | 56–75 | 9 |
| | 75–489 | 10 |
| Aspect | Flat | 1 |
| | North | 9 |
| | Northeast | 9 |
| | East | 5 |
| | Southeast | 5 |

**Table 4.** *Cont.*

| Factors | Classes | Suitability Rating |
|---|---|---|
| | South | 1 |
| | Southwest | 1 |
| | West | 1 |
| | Northwest | 5 |
| Slope (°) | 0 | 1 |
| | 0–1 | 2 |
| | 1–2 | 3 |
| | 2–3 | 4 |
| | 3–4 | 5 |
| | 4–6 | 6 |
| | 6–8 | 7 |
| | 8–10 | 8 |
| | 10–14 | 9 |
| | 14–61 | 10 |
| Curvature | Concave | 10 |
| | Flat | 1 |
| | Convex | 10 |
| TWI | 3–5 | 1 |
| | 5–6 | 2 |
| | 6–7 | 3 |
| | 7–8 | 4 |
| | 8–9 | 5 |
| | 9–10 | 6 |
| | 10–11 | 7 |
| | 11–12 | 8 |
| | 12–13 | 9 |
| | 13–17 | 10 |
| TRI | 1–2 | 1 |
| | 2–3 | 2 |
| | 3–4 | 3 |
| | 4–5 | 4 |
| | 5–6 | 5 |
| | 6–7 | 6 |
| | 7–8 | 7 |
| | 8–9 | 8 |

**Table 4.** *Cont.*

| Factors | Classes | Suitability Rating |
|---|---|---|
| SPI | −7 | 1 |
| | −1−−4 | 2 |
| | −4−−3 | 3 |
| | −3−−2 | 4 |
| | −2−−1 | 5 |
| | −1–0 | 6 |
| | 0–1 | 7 |
| | 1–2 | 8 |
| | 2–3 | 9 |
| | 3–4 | 10 |
| Agriculture (m) | 0 | 10 |
| | 0–53.7 | 9 |
| | 53.7–107.5 | 8 |
| | 107.5–161.2 | 7 |
| | 161.2–268.7 | 6 |
| | 268.7–376.2 | 5 |
| | 376.2–537.5 | 4 |
| | 537.5–752.5 | 3 |
| | 752.5–1128.8 | 2 |
| | 12,128–13,706 | 1 |
| Residential (m) | 0 | 1 |
| | 0–108.5 | 2 |
| | 108.5–217.1 | 3 |
| | 217.1–379.8 | 4 |
| | 379.8–542.7 | 5 |
| | 542.7–759.7 | 6 |
| | 759.7–976.8 | 7 |
| | 976.8–1302.5 | 8 |
| | 1302.5–1790.9 | 9 |
| | 1790.9–13,839.1 | 10 |
| Highway (m) | 0–1070.5 | 10 |
| | 1070.5–2542.4 | 9 |
| | 2542.4–4014.4 | 8 |
| | 4014.4–5486.3 | 7 |
| | 5486.3–6958.2 | 6 |
| | 6958.2–8564.1 | 5 |
| | 8564.1–10,303.6 | 4 |
| | 10,303.6–12,578.4 | 3 |
| | 12,578.4–15,254.7 | 2 |
| | 15,254.7–34,122.4 | 1 |

**Table 4.** *Cont.*

| Factors | Classes | Suitability Rating |
|---|---|---|
| Road (m) | 0 | 1 |
| | 0–110.2 | 2 |
| | 110.2–220.5 | 3 |
| | 220.5–330.8 | 4 |
| | 330.8–496.2 | 5 |
| | 496.2–661.6 | 6 |
| | 661.6–882.2 | 7 |
| | 882.2–1157.9 | 8 |
| | 1157.9–1654.1 | 9 |
| | 1654.1–14,060.6 | 10 |
| River (m) | 0 | 10 |
| | 0–1150.2 | 9 |
| | 1150.2–2091.4 | 8 |
| | 2091.4–2927.9 | 7 |
| | 2927.9–3764.5 | 6 |
| | 3764.5–4653.3 | 5 |
| | 4653.3–5699.1 | 4 |
| | 5699.1–6953.9 | 3 |
| | 6953.9–8993.1 | 2 |
| | 8993.1–13,332.7 | 1 |
| Population Density | 55–82 | 10 |
| | 82–110 | 9 |
| | 110–146 | 8 |
| | 146–241 | 7 |
| | 241–291 | 6 |
| | 291–412 | 5 |
| | 412–558 | 4 |
| | 558–1068 | 3 |
| | 1068–1678 | 2 |
| | 1678–4837 | 1 |
| Population | 500–2500 | 10 |
| | 2500–4800 | 9 |
| | 4800–5700 | 8 |
| | 5700–8300 | 7 |
| | 8300–8700 | 6 |
| | 8.700–11,300 | 5 |
| | 11,300–16,200 | 4 |
| | 16,200–21,300 | 3 |
| | 21,300–29,300 | 2 |
| | 29,300–64,400 | 1 |

Natural break, quantile, standard deviation, and equal interval are the most common techniques used by researchers to convert data into categorized formats. A quantile algorithm distributes a set of grid values into groups, each made up of an equal number of unique values assigned using the middle and extreme values. Quantile algorithms are designed so that each class of the conditioning factors is well-represented on the map for easy computation after it is converted to ordinal data [46]. For these advantages, the quantile classification method was used to classify the parameters.

### 3.3.1. Topographical Factors

Since digital elevation model (DEM) data are valuable instruments for suitability assessment of surface topographical and hydrological analysis, several terrain features are derivable for site suitability studies. The DEM has had an enormous impact on terrain analysis among environmental scientists, agricultural experts, geotechnical engineers, and hydrologists. Derived terrain factor reliability has been further enhanced by advances in data capture tools and GIS analysis capabilities [79]. The following subsections describe all the topographic conditioning factors extracted by the DEM.

### 3.3.2. Surface Elevation (Altitude)

Surface elevation has been vital in several suitability studies. A conditioning factor shows deviation of the maximum and minimum elevation of terrains [80]. Altitude controls microclimate factors, such as vegetation distribution, geomorphological characteristics, and surface runoff. High elevation combined with other terrain characteristics determines where facilities, such as hospitals, should be built due to accessibility and geohazard exposure.

### 3.3.3. Surface Slope

Surface slope is an important topographical factor often used in many terrain-related studies, especially in hydrology, due to its effects on surface runoff accumulation and velocity of excess rainfall [72]. Slope regulates the direction of water flow and the speed and extent to which it spreads. It also impacts vehicular movement, particularly at steep slopes.

### 3.3.4. Surface Aspect

Aspect is an essential factor that also influences hospital site selection. It reveals the direction the slope faces. Aspect affects the microclimate of the slope, as it bonds to solar insulation and wind moisture content, etc. Hillsides fronting the sunshine usually provide better illumination than those facing away. It equally determines the intensity and pattern of rainfall, lineament, and wind effects.

### 3.3.5. Surface Curvature

Curvature measures the rate of change of a slope characterized as profile and planimetric (planar) curvature [81]. Curvature is usually coded as zero, negative, and positive on a raster map, where positive values describe a convex landscape, negative values describe concave, and pixels coded zero describe flat terrain. Concavity and convexity show high water runoff and accumulation areas, respectively, while flat terrain favors water stagnation, indicating a susceptibility to flooding [46].

### *3.4. Hydrological Indices (SPI, TWI, TRI)*

*SPI* and *TWI* are precipitation-related topographic factors obtained by applying Equations (1) and (2), respectively [82]:

$$SPI = As\tan\beta \tag{1}$$

$$TWI = In\left(\frac{As}{tan\beta}\right) \tag{2}$$

where *As* represents the catchment area or flow accumulation (m$^2$ m$^{-1}$) and $\beta$ is the local slope gradient measured in degrees.

Flow algorithms are a vital component of hydrological and terrain analysis and are particularly relevant in this case, since surface flow determines the amount and concentration of surface and sub-surface runoff, which, together with slope, control the stability of any location. Several hydrological parameters are derivable from the DEM, such as flow direction and flow accumulation, and can be generated using ArcGIS software. valign="middle" indicates the erosive power of water flow [83], while *TWI* characterizes the effects of topography on runoff generation and the amount of flow accumulation at any location within a river catchment [84].

*TRI* is another morphological parameter widely used in flood analysis, which can be calculated using Equation (3):

$$TRI = \sqrt{Abs(max^2 - min^2)} \tag{3}$$

where *max* and *min* show the most significant and most negligible values, respectively, of cells in nine rectangular neighborhoods of altitude.

### 3.5. Environmental Factors

The land use data used in this study were essentially the available data with precise classifications for a particular area. In this study, land use data were derived from the vector map using an ArcGIS 10.5 analysis of Euclidean distance. Then, the derived conditioning factor shapefiles were converted into a 10 m raster map for each category and divided into ten classes using the quantile method.

#### 3.5.1. Distance from River Network

Distance from rivers was regarded as another relevant conditioning factor. With increased runoff and discharge into rivers and streams, there may be a chance of flooding in regions adjacent to rivers, especially at descending elevations and slopes. In such cases, the interval of a possible location from rivers and streams is essential to incorporate into hospital site suitability criteria.

#### 3.5.2. Distance from Highway and Road

As proximity to rivers was an important parameter to consider, the distance from roads was also critical. Accessibility is significant to locate a new hospital, so decision makers are sometimes concerned with locations near roads, especially major roads; the closer the hospital is to the road, the easier to provide services and maintain the facility. In addition to the advantages of proximity to roads, there are also concerns about car or train noise pollution. For this reason, there is always a middle ground between how close or far the ideal site is located.

#### 3.5.3. Distance from an Agricultural Area

For the well-being of any society, food security is fundamental. Policies exist worldwide to prohibit the location of agricultural grounds. Therefore, a distance from agricultural areas was enforced to evaluate the suitability of land for hospital buildings.

#### 3.5.4. Distance from the Residential Area

Distance from residential areas was also regarded as a relevant conditioning factor. The closer healthcare facilities are to urban areas, the more convenient they are. Residential parameters are usually scanned based on population and distribution data converted to an objective map layer in the form of bitmaps. The more details available, the more acceptable the overall accuracy of the resulting factor.

*3.6. Geodemographic Factors*

3.6.1. Population Factors

The location of a hospital close to an urban area directly influences emergency response in times of disaster for cost-benefit emergency service in the event of emergencies. It is essential to know the population dynamics, such as size, density, age, and gender in the neighborhoods of potential hospital sites. For this reason, population size and density were selected as conditioning factors in this study. The population data were created using a population census and the vector data of every area in ArcGIS 10.5.

3.6.2. Population Size

Population size is related to hospital demand and performance. A population census was collected and divided according to each local governorate and district. The population data of Malacca were collected from Jabatan Perangkaan Malaysia for 2019.

3.6.3. Population Density

Population density is also linked with hospital prospective demand and significant performance. Population density estimates the number of people in a measured unit area as a function of the values of individual kilometers or per square meter, estimated using census data on the base of census tracts as a spatial unit for analysis. Population density was measured using Equation (4).

$$Population\ density = \frac{Number\ of\ people}{Land\ area} \tag{4}$$

In terms of suitability, the conditioning factors were reclassified into different suitability levels to be weighed according to the results of both AHP and MLP. Table 4 shows the suitability rating of each conditioning factor.

*3.7. Methodology*

3.7.1. Overview

Eight hospital inventory sampling datasets were acquired from Jabatan Kesihatan Negeri Melaka, and an additional eight non-hospital locations (as the dependent variables) were selected randomly and counted to the hospital pinpoints. Then, the extractions of 14 factors (as independent variables) were obtained from different authorities. The conditioning factor values associated with the sampling point positions were collected as modeling process inputs.

AHP modeling was carried out through weighting and pairwise comparison of the input variables (conditioning factors). Operationally, pairwise comparison forms judgments between pairs of the set of factors, rather than prioritizing the elements [85]. The AHP procedures comprise seven steps: (1) constructing a pairwise comparison, (2) normalizing the matrix weight, (3) deriving the priority sector, (4) calculating the maximum Eigenvalue, (5) computing the consistency index (CI) using Eigenvalue, (6) computing the random index (RI), and (7) calculating the consistency ratio (CR). Note that the value of the CR obtained must be lower than 0.1 (10%) for the results to be acceptable [46].

For the MLP modeling operation in the Waikato Environment for Knowledge Analysis (Weka), the sampling data, comprising a combination of independent and dependent variables, was split into 70% training and 30% validation datasets. The outcomes were consecutively employed to create the maps of hospital site suitability. The implementation of both approaches was assessed utilizing the test sample data set with statistical evaluation metrics and ROC curves. Figure 4 illustrates the general workflow of this paper.

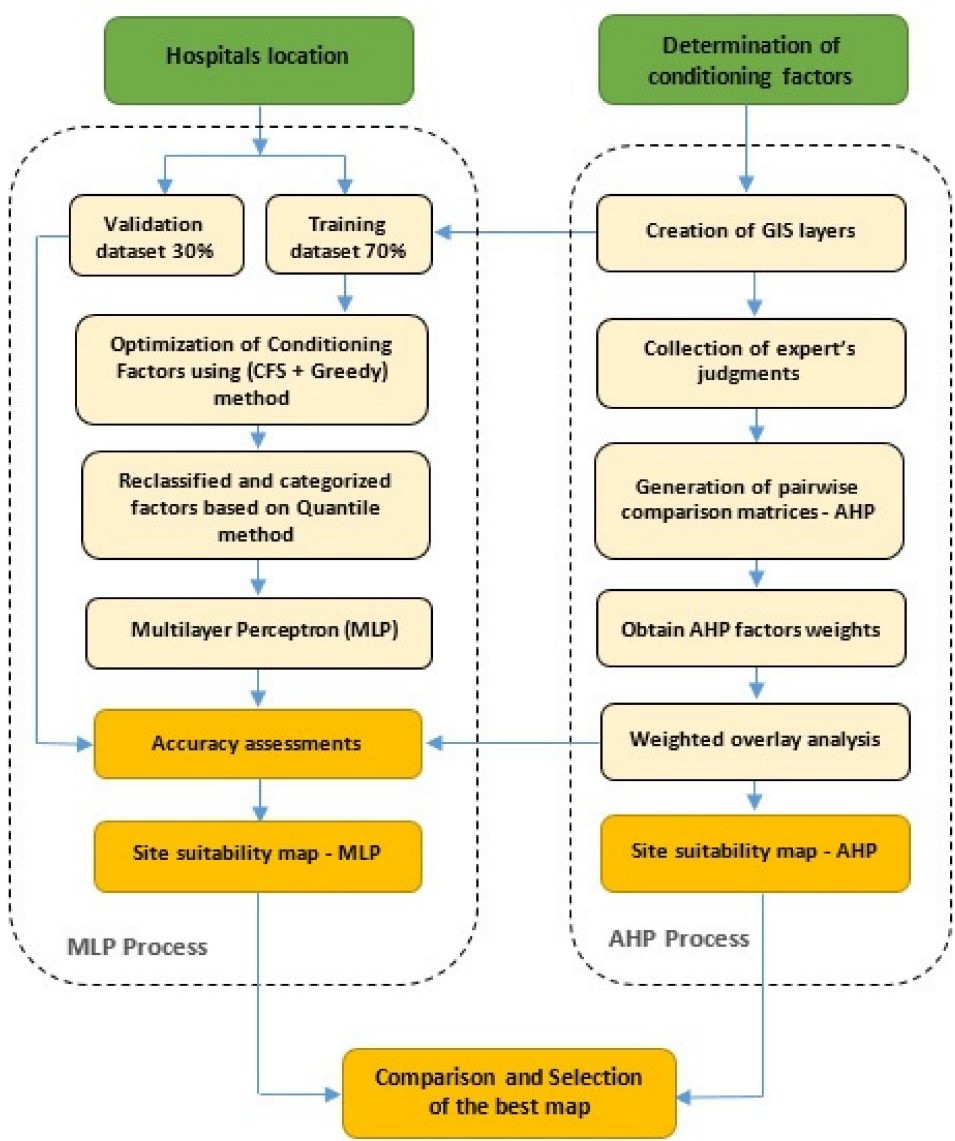

**Figure 4.** General methodological flowchart.

3.7.2. Factor Analysis

An exploration of filter-based feature selection algorithms and classifiers to increase learning algorithm performance in high-dimensional datasets was carried out in this study. A filter-based feature selection algorithm, correlation-based feature selection (CFS), was applied for the best feature selection depending on the relevance of the criterion. CFS was cited as the best selection feature that improved the quality of topographic and hydrological factors, as it consists of two steps: ranking the initial features and removing less significant features through an iterative process [86]. CFS utilizes appropriate correlation measures and a heuristic search strategy [87]. Many authors have studied the heuristic and greedy stepwise search strategies, and they have been reported to be the best methods [88]. Here, a greedy stepwise search strategy was used to distinguish stable features from the pool of features. The greedy stepwise filter feature selection technique was more suitable for criterion selection in this case because it is robust, and it is easy to interpret the stability of a selected feature. Vanaja and Kumar [89] defined the advantages of this search methods as:

- Allows the learning algorithm to train faster;
- Minimizes ambiguity of a model and makes it easier to analyze;
- Improves the performance of the learner;

- Eliminates redundancy.

Because of the highlighted advantages above, a CFS filter with greedy stepwise search techniques was used to optimize and rank the significance of the conditioning factors according to their influence based on the location of the hospital. Through this, model performance was strengthened by the exclusion of irrelevant factors.

### 3.8. AHP

AHP is a widely employed multicriteria method for analyzing complicated decisions where parameters are structured in a hierarchical order. This methodology considers the relative significance of the factors according to the AHP scale. The stated scale directs urban planners and decision makers to create a pairwise comparison matrix (PCM). The pairwise comparison matrix reveals the possibility of each element to be equally strong, slightly firm, reasonably strong, very strong, or more robust than the other factors. These relative intensities are further converted into numbers, (shown in Table 5). Furthermore, various procedures are available to resolve the preference vector W. In a consistent matrix, each procedure provides a true preference vector V. Depending on the decision problem size and the accuracy of expert assessment and, thus, the inconsistency degree of pairwise comparison matrices, individual procedures allow for the determination of a different estimation of the vector V [90]. In this study, a normalized column sum was used to calculate the PCM.

**Table 5.** Scale for pairwise comparison.

| Intensity of Importance | Definition |
|:---:|:---:|
| 1 | Equal importance |
| 2 | Equal to moderate |
| 3 | Moderate to importance |
| 4 | Moderate to strong importance |
| 5 | Strong importance |
| 6 | Strong to very strong importance |
| 7 | Very strong importance |
| 8 | Very to extremely strong importance |
| 9 | Extreme importance |

Source: [91].

The following steps were applied to classify the weight of the criteria during the AHP process:

1. Development of a pairwise comparison matrix.

The method used a scale with values ranging from 1–9 (Table 5) to rate the relative preferences for each pair of criteria evaluated.

2. Computation of criterion weight.

Computation of criteria weights involved three operations: First, the values were summed in each column of the matrix and then each element was divided by its column total (the resulting matrix is referred to as the normalized pairwise comparison matrix). After that, the average of the elements in each normalized row was computed by dividing the sum of normalized values for each row by the number of criteria. The resulting standards provided an estimate and compared the relative weights of the criteria. The next steps were followed to calculate the final weights for all factors [92].

Sum the pairwise comparison matrix values in every column by the following formula:

$$L_{ij} = \sum_{n=1}^{n} C_{ij} \tag{5}$$

where $L_{ij}$ represents the pairwise comparison matrix total column value and $C_{ij}$ represents the criteria applied for the analysis.

Divide every component in the matrix by its total row to obtain a normalized pairwise comparison matrix:

$$X_{ij} = \frac{C_{ij}}{\sum_{n=1}^{n} C_{ij}} \tag{6}$$

where $X_{ij}$ is the normalized pairwise comparison matrix.

Divide the sum of the matrix's normalized row by the number of the parameter ($N$) to obtain the standard weight by applying the following formula:

$$W_{ij} = \frac{\sum_{j=1}^{n} X_{ij}}{N} \tag{7}$$

where $W_{ij}$ is the standard weight.

To calculate the consistency vector values, the following formula was applied:

$$\lambda = \sum_{i=1}^{n} CV_{ij} \tag{8}$$

where $\lambda$ is the consistency vector.

The weighted sum vector was first calculated by multiplying the pairwise matrix with the weight to approximate consistency and then followed by the division of individual elements by their corresponding weights, i.e., the first element divided by its weight, after which the consistency vector was created by dividing the weighted sum by the weights. The lambda and CI were then computed from the consistency vector. Lambda is the mean of the consistency vector, while the CI is derived according to an inspection that must be equivalent to or higher than the numeral of parameters. A measure of inconsistency degree was computed by subtracting the number of parameters from lambda ($\lambda$-$n$).

3. Estimation of the CR.

The CR is important for establishing whether the comparisons are consistent or not. The process of estimating CR involved a few steps: First, the weighted sum vector was determined by multiplying the weight of the first criterion by the initial column of the original pairwise comparison matrix, and so on, until the last criterion was multiplied by the previous column. Then, the values were summed over the rows. Second, the consistency vector was created by dividing the weighted sum vector by the criterion weights. Third, lambda ($\lambda$), the average value of the consistency vector, and CI (Equation (9)), which measures the departure from consistency, were computed.

$$CI = (\lambda - n)(n - 1) \tag{9}$$

Lastly, the CR was obtained using Equation (10):

$$CR = CI/RI \tag{10}$$

where RI is the random index. RI depends on the number of elements being compared (Table 6). A CR value of < 0.10 means that the pairwise comparison is consistent and acceptable, but if the CR value is $\geq$ 0.10, the pairwise comparison is inconsistent, and the entire process must be revised.

**Table 6.** Random inconsistency indices (RI).

| n | RI |
|---|---|
| 1 | 0.00 |
| 2 | 0.00 |
| 3 | 0.58 |
| 4 | 0.90 |
| 5 | 1.12 |
| 6 | 1.24 |
| 7 | 1.32 |
| 8 | 1.41 |
| 9 | 1.45 |
| 10 | 1.49 |
| 11 | 1.51 |
| 12 | 1.48 |
| 13 | 1.56 |
| 14 | 1.57 |
| 15 | 1.59 |

Source: [91].

The identification and selection of factors that impact achieving the best output are essential. AHP provides weights and ranks the conditioning factors according to how well they predict the best location. In this study, AHP was executed to weigh the conditioning factors in the Expert Choice package. The steps involved were:

Creating a pairwise matrix: an evaluation scale of 1 to 9 was assigned to the parameters to rate the relative importance of each pair of input data layers. For example, if alternative A has double preference over B, then B has half priority compared to A, but a comparison of each criterion alone will result in a score of 1, which is interpreted to mean equivalent priority. Therefore, in the pairwise comparison matrix, all the diagonal elements were 1.

Criteria weighting: this stage included:

a. Summing up the values of each column in the pairwise matrix;
b. Dividing the matrix element by its column total (to derive the normalized matrix);
c. Calculating the average of the elements in every row of the normalized matrix to obtain an estimated relative priority of the elements being compared.

Table 7 presents the matrix of the relative importance of the conditioning factors.

**Table 7.** The fundamental assessment scale for the AHP.

| Description | Intensity of Importance |
|---|---|
| Extremely less important | 1/9 |
| | 1/8 |
| Very strongly less important | 1/7 |
| | 1/6 |
| Strongly less important | 1/5 |
| | 1/4 |
| Moderately less important | 1/3 |
| | 1/2 |
| Equal importance | 1 |

**Table 7.** *Cont.*

| Description | Intensity of Importance |
|---|---|
| | 2 |
| Moderately more important | 3 |
| | 4 |
| Strongly more important | 5 |
| | 6 |
| Very strongly more important | 7 |
| | 8 |
| Extremely more important | 9 |

Source: [93].

Assessing the consistency matrix: this process involved:

a. Determining the total weighted vector. To achieve this, the weight of the first scale was multiplied by the first column of the leading binary comparative matrix and then multiplied by the second scale of the second column. Then, the third scale was multiplied by the third column of the primary matrix, and, finally, these values were summed;

b. Determining the consistency vector: we divided the weight vector by the scale weights. Using the weight produced by AHP, the conditioning factors were combined in an ArcGIS environment using the weighted sum overlay tool to create a final suitability map. Table 8 presents the pairwise comparison matrix of the selected conditioning factors for hospital site suitability in Malacca.

**Table 8.** Pairwise comparison matrix (Malacca): (A) residential, (B) road, (C) agriculture, (D) highway, (E) population density, (F) population, (G) curvature, (H) aspect, (I) *SPI*, (J) slope, (K) altitude, (L) river, (M) *TWI*, and (N) *TRI*.

| | A | B | C | D | E | F | G | H | I | J | K | L | M | N |
|---|---|---|---|---|---|---|---|---|---|---|---|---|---|---|
| **A** | 1 | 2 | 2 | 1 | 4 | 5 | 2 | 7 | 8 | 9 | 7 | 6 | 5 | 3 |
| **B** | 1/2 | 1 | 3 | 3 | 4 | 6 | 6 | 4 | 6 | 6 | 6 | 4 | 3 | 4 |
| **C** | 1/2 | 1/3 | 1 | 2 | 2 | 5 | 4 | 3 | 6 | 7 | 6 | 2 | 5 | 3 |
| **D** | 1 | 1/3 | 1/2 | 1 | 1 | 4 | 3 | 3 | 1 | 5 | 4 | 1 | 2 | 8 |
| **E** | 1/4 | 1/4 | 1/2 | 1 | 1 | 3 | 2 | 1/2 | 5 | 3 | 5 | 1 | 2 | 6 |
| **F** | 1/5 | 1/6 | 1/5 | 1/4 | 1/3 | 1 | 1/3 | 1/3 | 1 | 3 | 4 | 1 | 3 | 7 |
| **G** | 1/2 | 1/6 | 1/4 | 1/3 | 1/2 | 3 | 1 | 1 | 4 | 6 | 5 | 1 | 3 | 1 |
| **H** | 1/9 | 1/8 | 1/7 | 1/5 | 1/5 | 1/3 | 1/6 | 1 | 1/2 | 6 | 1 | 1 | 2 | 4 |
| **I** | 1/2 | 1/2 | 1/2 | 1/5 | 1/7 | 1/7 | 1/5 | 1/5 | 1 | 7 | 2 | 2 | 2 | 3 |
| **J** | 1/9 | 1/6 | 1/7 | 1/5 | 1/3 | 1/3 | 1/6 | 1/6 | 1/7 | 1 | 1/2 | 1/7 | 4 | 2 |
| **K** | 1/7 | 1/6 | 1/6 | 1/4 | 1/5 | 1/4 | 1/5 | 1/2 | 1/2 | 1/2 | 1 | 1/9 | 2 | 2 |
| **L** | 1/6 | 1/4 | 1/2 | 1 | 1/3 | 1/2 | 1/2 | 1/2 | 1/2 | 1/3 | 1/3 | 1 | 9 | 3 |
| **M** | 1/2 | 1/3 | 1/4 | 1/3 | 1/9 | 1/9 | 1/5 | 1/6 | 1/5 | 1/7 | 1/2 | 1/7 | 1 | 5 |
| **N** | 1/9 | 1/4 | 1/3 | 1/3 | 1/5 | 1/9 | 1/3 | 1/2 | 1/7 | 1/2 | 1/2 | 1/6 | 1/5 | 1 |

*3.9. ML Model*

This study also implemented an MLP model in WEKA 3.8.2, which was designed by the University of Waikato, New Zealand [94]. The sample data was subdivided into

70% training and 30% validation subsets by applying Weka's random partitioning algorithm. Technically, there is no typically accepted technique to segment a sampling data set; selection differs according to the literature and usually depends on the quantity and quality of sample data [74]. In the pre-data modeling stage, the data were subjected to CFS and a greedy stepwise search algorithm with a 10-fold cross-validation technique to minimize failure, improve effectiveness, and reach adequate implementation [95,96]. CFS is a promising feature selection method; the greedy stepwise algorithm counts either the foremost satisfactory feature or eliminates the foremost unsuitable feature in every round [97]. In this study, the procedure ranked parameters according to their impact on the model. Both models were analyzed and performed in ArcGIS 10.5. Using quantile classification, the outputs were reclassified into five suitability classes to create the output maps [46]. The quantile classification process measured the equivalent illumination of every class by statistically estimating the raster value ranges in the inputs [98].

Selecting suitable ML classification algorithms is vital for accuracy improvement [99,100]. The two main classification algorithms are supervised and unsupervised [62,101]. MLP is a promising supervised classification for hospital site suitability [19].

MLP has three different layers (Figure 5): the input layer, which transfers the input vector to the network; the hidden layers that contain the computation neurons; and the output layer, which is made up of at least a computation neuron that produces the output vector. Many perceptrons link the layers, and each connection has associated weight. The work of the hidden layers is identical to the kernel function in SVM, which is projecting the feature vectors to high-dimensional space to find a hyper plain that separates the training data properly. In MLP, each layer creates a feature vector for the input data based on the assigned weight value. Features from one layer are transferred onto the next via a non-linear initiation operation that alters the feature space to project the data as input into the next layer to be mapped to a new feature space [49,102].

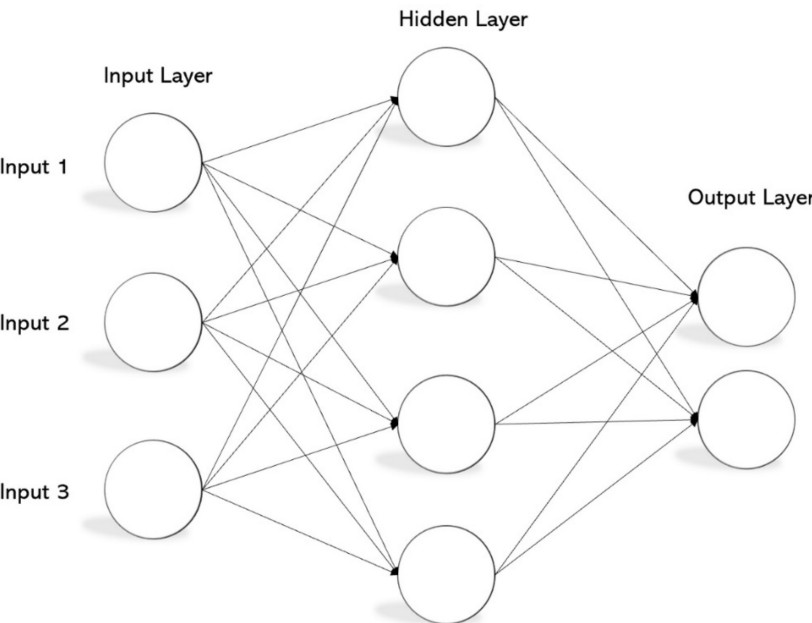

**Figure 5.** MLP layers.

A perceptron is a mathematical model similar to a biological neuron transmission, similar to the propagation of electrochemical stimulation in a neuron [103]. During transmission, only the filtered information via activation rule (non-linear function) is passed to the next perceptron, a process called feed-forward computation [81]. Binary or bipolar sigmoid activation functions are often preferred [79].

The main objective of a training task in MLP is to discover an appropriate weight set that will reduce cost function, preferably to 0. Generally, the cost function estimates the difference between the predicted and actual values. For every iteration, all the weights in the network are updated to minimize the output of the cost function. The mechanism of deciding the degree of the weights is known as a learning algorithm, and gradient descent is commonly used by researchers. The receiving weight is adjusted to be equal to the learning rate depending on the quality of the cost function, but in the opposite direction of the cost function. When a lesser learning rate is used, the cost function requires considerable time and computation tasks to reach a minimum value, but the reverse is the case with a significant learning rate [79].

According to [104], the two historically standard activation functions are sigmoid, as expressed in Equation (11):

$$y(v_i) = \left(1 + e^{-v_i}\right)^{-1} \tag{11}$$

Equation (11) is called a logistic activation function, and the value ranges from 0 to 1. Note that $y_i$ in the $i^{th}$ is the neuron output that accepts the weighted sum of input connections $v_i$. The error degree in output node $j$ in the $n^{th} the$ data point is obtained through:

$$e_j(n) = d_j(n) - y_j(n) \tag{12}$$

where $d$ and $y$ are the target and produced value by the perceptron, respectively. Based on the error, node weights are updated to minimize error in the output using the expression in Equation (13):

$$\epsilon(n) = \frac{1}{2} \sum_j e_j^2(n) \tag{13}$$

and the weight is updated using Equation (14):

$$\Delta w_{ij}(n) = -\eta \frac{\partial \epsilon(n)}{\partial v_j(n)} y_i(n) \tag{14}$$

where $y_i$ is the output and $\eta$ is the learning rate, which is simplified using Equation (15):

$$-\frac{\partial \epsilon(n)}{\partial v_j(n)} = e_j(n) \varnothing'(v_j(n)) \tag{15}$$

where $\varnothing'$ is the derivative activation function. The change in weight in the hidden node can be effectively computed using Equation (16):

$$-\frac{\partial \epsilon(n)}{\partial v_j(n)} = \varnothing'(v_j(n)) \sum_k -\frac{\partial \epsilon(n)}{\partial v_k(n)} \left(w_{kj}(n)\right) \tag{16}$$

The error depends on the change in weights backward from the output layer, the hidden layer, and, finally, to the input layer—backpropagation [105].

### 3.10. Validation

Validation is an important part of any predictive modeling. In this analysis, the data were subdivided into two different data sets: 70% training, and 30% validation. For the MLP model, 10-fold cross-validation was performed. The process of cross-validation divided the data set into ten subsets; nine subsets were used for training and the residual subset for testing. Individually, in each round of the 10-fold process, a distinct component was used to test the accuracy, and the last outcome signified an average of the ten outcomes [106]. Model performance was evaluated using the correlation coefficient (R2), and the statistical evaluation metrics were root mean square error (RMSE), root relative squared error (RRSE), relative absolute error (RAE), and mean absolute error (MAE) in the cross-validation test. Model implementation was also validated using the receiver operating characteristic (ROC).

## 4. Results

The AHP method was used to weigh the conditioning factors (Table 9). These factors were also overlayed using a weighted sum algorithm to identify potentially suitable areas to build a hospital. Table 9 shows the AHP weight for the Malacca case study dataset. Mainly, distance to residential area contributed significantly to assessing hospital site suitability by 22%, and due to its influence, it was ranked the best variable. Following narrowly in the contribution rank were roads (21%), distance from agricultural areas (14%), and highways (10%). The next group comprised population density and curvature, which made up 7% each, and *SPI* (5%). Least in the rank were population and aspect (3%), altitude, *TRI*, and *TWI* (2%), and the least contributing factor, slope, had just one. Also, the AHP pairwise comparison matrix consistency results produced a promising agreement with a CR value of 0.08, which is within the acceptable limit.

**Table 9.** AHP conditioning factor weights (Malacca).

| Conditioning Factors | Weights |
|:---:|:---:|
| Population Density | 0.074 |
| Population | 0.034 |
| Altitude | 0.015 |
| Agriculture | 0.136 |
| Residential | 0.218 |
| Road | 0.208 |
| Highway | 0.096 |
| River | 0.03 |
| Slope | 0.012 |
| Curvature | 0.066 |
| Aspect | 0.027 |
| *TRI* | 0.016 |
| *TWI* | 0.02 |
| *SPI* | 0.048 |

The AHP-weighted overlay hospital site suitability map of Malacca (Figure 6) has five classes of suitability distributed almost equally in the coverage area (i.e., an average of 20%). For instance, 19.5% of the site was classified as very highly suitable, 22% was high suitability class, and 19.7%, 20.5%, and 18.3% were shared among the moderate, low, and very low suitability classes, respectively. The best two classes, the very high and high suitability classes, were about 41.5% of the total areas common in a highly populated urban area in the three districts of Malacca, and the other less suitable classes were located in less densely populated areas. The distribution of the suitability classes was proportional to conditioning factors, such as residential areas, highways, agriculture areas, roads, population density, altitude, slope, and *SPI*. The Malacca hospital site suitability map shows similarity with the map produced using the MLP model.

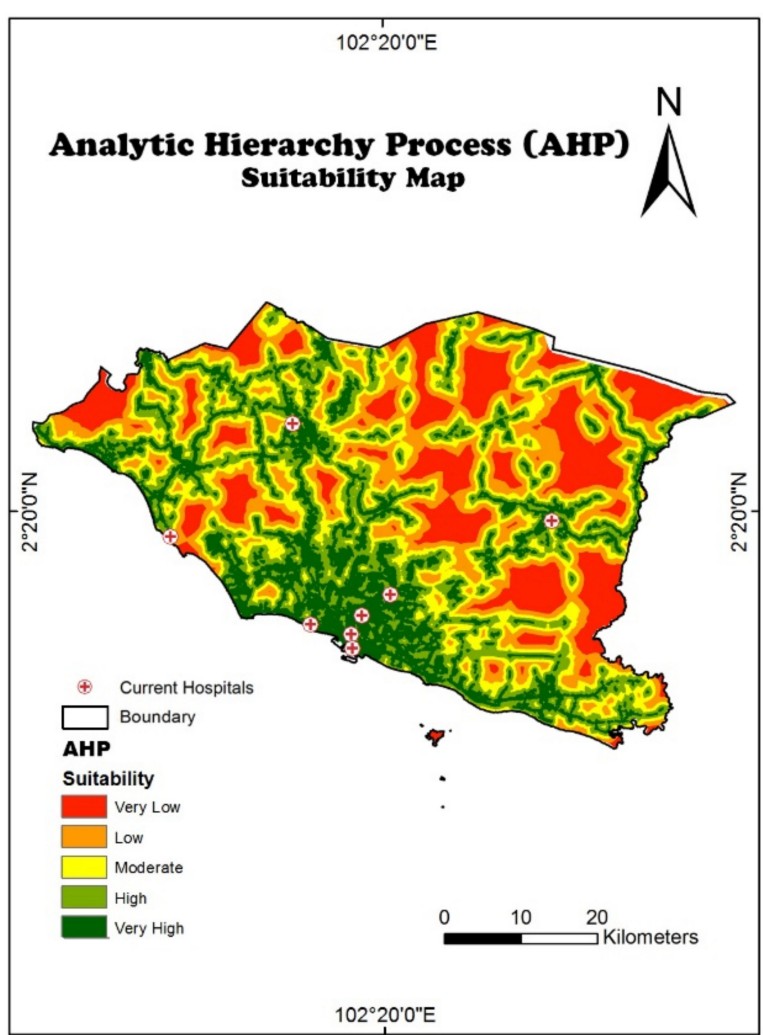

**Figure 6.** Suitability map produced using Analytical Hierarchy Process (AHP).

The suitability maps produced with the AHP-created weights are presented in Figure 6. The AHP performance was validated using a pairwise comparison matrix (consistency test) delivering a CR value of 0.083. The result is rather suitable and reliable since the CR value was less than the CR < 0.1 threshold.

Evaluation of hospital location-related conditioning factors using CFS provided insight into the significance level of conditioning factors. The CFS ranked the conditioning factors based on class label correlation and other factors (Table 10). It was deduced from Table 10 that the relative impact between hospital sites and other factors indicated that population density had the highest influence (100%), followed by distance from the road, distance from agriculture, distance from residential areas (90%), distance from the highway, distance from the river, and population number (80%). The factors in the mid-range were slope, altitude, *TRI* and *TWI* (70%), SPI, and plan curvature with a relative influence value of 60% and an aspect of 40%. The CFS was evaluated on merit and was evaluated using a correlation coefficient and error rates.

**Table 10.** The relative influence of the conditioning factors.

| Parameters | Values | Relative Influence % |
|---|---|---|
| Population density | Density of population | 100% |
| Road | Distance from the road | 90% |
| Agriculture | Distance from agriculture | 90% |
| Residential | Distance from residential areas | 90% |
| Highway | Distance from the highway | 80% |
| River | Distance from the river | 80% |
| Population | Population number | 80% |
| Slope | Slope degree | 70% |
| Altitude | Altitude | 70% |
| *TRI* | Topographic roughness index | 70% |
| *TWI* | Topographic wetness index | 70% |
| *SPI* | Stream power index | 60% |
| Curvature | Plan curvature | 60% |
| Aspect | Aspect | 40% |

In this study, we analyzed the MLP model performance at different levels of processing. Initially, the feature selection level evaluated the conditioning factors associated with the models by analyzing cross-validation accuracy measurements (correlation coefficient, RMSE, RRSE, RAE, and MAE). Also, the implementation of the model and specification utilizing the metrics of the ROC curves were evaluated. Table 11 shows the outcome for both evaluation stages.

**Table 11.** Model validation results: area under the curve and CFS 10-fold cross-correlation.

| | **95% Confidence Interval** | | | |
|---|---|---|---|---|
| **Model** | **AUC** | **Std. Error** | **Lower Bound** | **Upper Bound** |
| MLP | 0.922 | 0.066 | 0.793 | 1.000 |
| AHP | 0.914 | 0.070 | 0.777 | 1.000 |
| | **10-Fold Cross-Correlation Method** | | | |
| | R2 | RMSE | RRSE (%) | RAE (%) | MAE |
| MLP | 0.998 | 0.0027 | 0.54 | 0.38 | 0.0019 |

The hospital site suitability map is shown in Figure 7. The map was created according to the influence of the conditioning parameters to determine potentially suitable locations for hospital construction in the study area. The models exploited the inter-relationships between the conditioning factors based on the sampled dataset to designate a relationship that ideally created the map of suitable hospital sites using the MLP model. The suitability map was classified into five classes from very low to very high (Figure 7) using the quantile classification method [46]. Quantitative assessment revealed that, for the MLP model, the suitability classes of the study area were distributed as very low (18.1%), low (20.2%), moderate (20.5%), high (20%), and very high (21.2%).

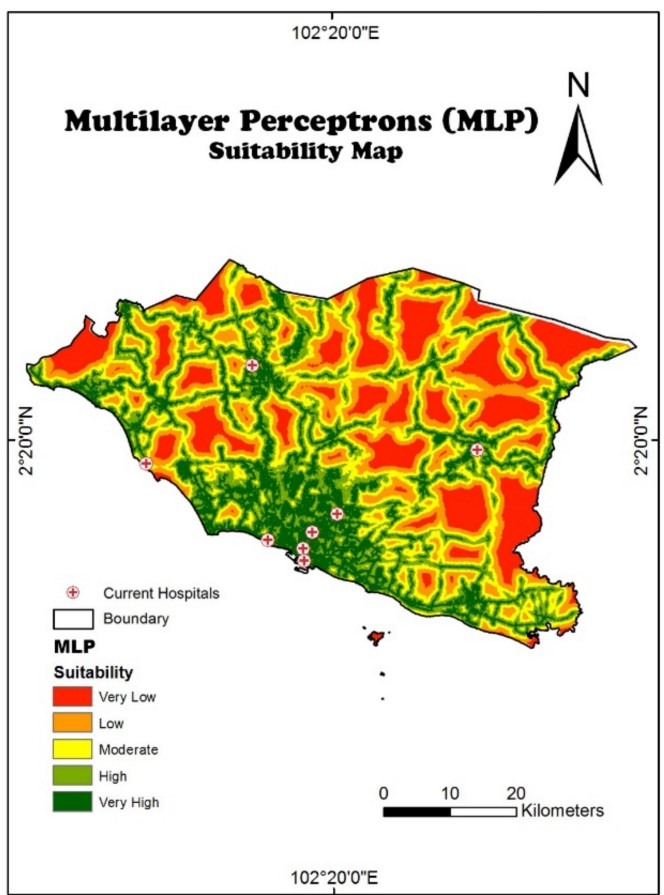

**Figure 7.** Suitability map produced using multilayer perceptron (MLP).

The cross-validation results of the model showed a high correlation between the conditioning factors and the location of the hospitals (Table 11). The MLP model had a very high correlation with an $R^2$ value of 0.99. The MLP model also had a low error rate, with RMSE, RRSE, RAE, and MAE values of 0.0027, 0.54%, 0.38%, and 0.0019, respectively. The cross-correlation was a pre-modeling estimation of the practicability of the MLP model for proper location forecasting with consideration to the conditioning factors.

Figure 8 and Table 11 present the performance of the model assessment using the validation dataset. The sensitivity-specificity indicators present a graphical and statistical knowledge of the efficiency of the models in distinguishing non-suitable locations from suitable locations for hospitals. It is noticeable in Figure 8 that the model curves converge upwards in the left corner of the plot, indicating that the outcomes have high overall accuracy [107]. Relatively, the modeling process outcomes had high classification ability with sensitivity and specificity values of 0.79 and 1.00 for the MLP model (Table 11). The model performance based on the analysis of ROC curves provided an overall accuracy value of 92%, including a standard error of 0.066 at a 95% confidence interval.

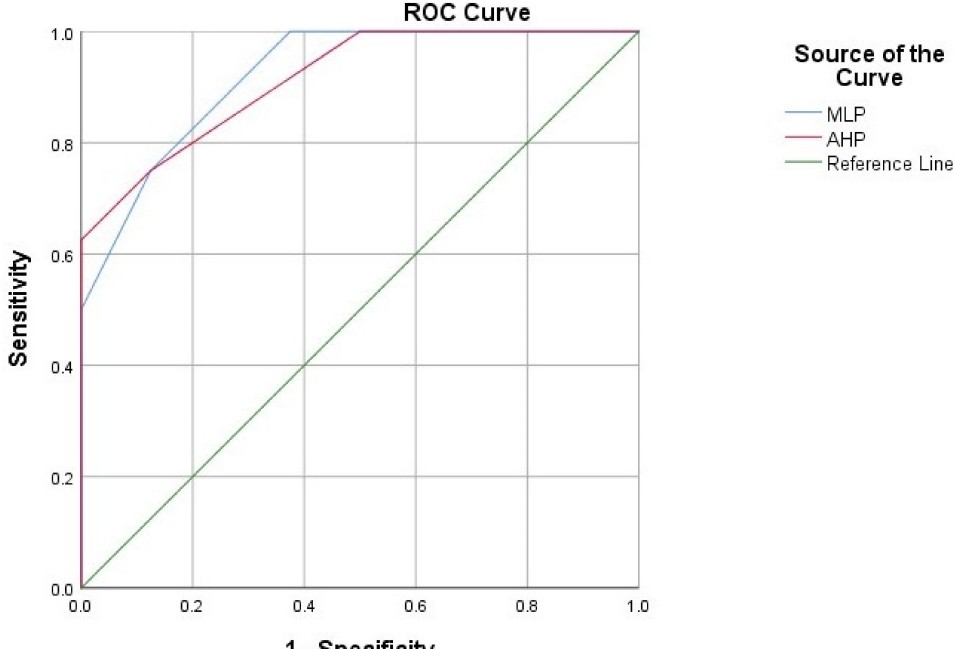

**Figure 8.** ROC curves of MLP and AHP.

## 5. Discussions

The AHP model was developed based on a pairwise matrix (weighted conditioning factors), and the model results were applied to the conditioning factors by weighted overlay (sum operator). AHP-weighted overlay uses discrete boundaries based on crisp sets, where each cell is either in a class or not. The AHP map discreet boundaries for suitability classification in the Malacca case study is clear.

The AHP and ML models were applied using the same weighted overlay tool. Compared to the other models, the AHP method is comprehensive and frequently used for hospital site suitability. AHP and MLP are more efficient for handling challenging and non-linear datasets. Both models are reliable, although they have a few differences in function and procedure. The AHP and MLP models produced good and reliable hospital site suitability studies, regardless of functional and practical differences.

According to Table 10, the relative influence of the conditioning factors indicated that population density had correlation values of 100%, followed by distance from the road, distance from the agriculture, and distance from residential areas (90%); and distance from the highway, distance from the river, and population number (80%). The factors in the mid-range were slope, altitude, *TRI*, and *TWI* (70%), as well as *SPI* and plan curvature, with a relative influence value of 60%, and an aspect of 40%. This revealed that demographic and topographic factors have a commanding influence on site selection.

Identification of suitable sites for hospital construction is a complex process that requires decision-making on the right conditioning factors considering their spatial heterogeneity. Accordingly, the first step to ensure a reliable outcome is to evaluate the relationships of the selected factors considered for choosing possible hospital sites by studying their correlation with models using the cross-correlation method. The correlation value obtained was 0.998 for the MLP model, indicating that the factors selected for the purpose were suitable.

The evaluation of model performance depends on the sensitivity and specificity of the ROC curves. The estimation of the curves was on a typical scale of 0 to 1, where a value less than 0.6 shows low accuracy while values between 0.6–0.7, 0.7–0.8, 0.8–0.9, and more than 0.9 are indicated to be within the medium, good, and very good accuracy ranges, respectively. [74]. Interactively, sensitivity and specificity measures provide an

understanding of model classification efficiency. Sensitivity, also known as true positivity, shows model ability to categorize data from samples to determine the ideal placement for constructing a hospital. In contrast, specificity, known as true negative, shows model ability to categorize data from samples to determine unsuitable sites. The acquired AUC values for the MLP and AHP models were 0.92 and 0.91, respectively. According to [107], on a scale of 0 to 1, the closer the value acquired is to 1, the better model ability of data classification. The evaluation performance of this study provided high accuracy results and demonstrated the superiority of the MLP model.

Based on author knowledge of the area of study integrated with the exploration of a Google Earth high-resolution map, we noticed that the MLP outcome reflected the environment of the study area. For example, very high and high suitability classes occurred in accessible and densely populated areas (Figure 7). Similarly, very low and low suitability classes were labeled as uninhabited and sparsely populated areas. Similarly, the AHP outcome (Figure 6) showed almost an identical classification with MLP.

The performance of the MLP and AHP methods in this study provides scientific evidence for the effectiveness of the ML method for assessing the suitability of a hospital site similar to findings in other study areas, such as floods [108,109], landslides [48,110], forest fires [111,112], erosion and water resources [113–115], etc. MLP and AHP performed satisfactorily in this investigation with AUC values of 92% and 91%, respectively. The MLP model is a suitable and convenient model due to its capability to construct and weigh the influencing criteria employing non-linear projection. The MLP model, for each subset of the training samples, computed the results of neurons from each layer and predicted the final output layer (forward pass). The prediction depended on the speed of calculating the variance between the expected and actual results to obtain the prediction error, which was subsequently run to change the weightiness of neurons in all previous layers (backpropagation) until achieving superior prediction precision [116,117]. This sophisticated process allowed it to accurately handle linear and non-linear data sets.

## 6. Conclusions

Hospitals are considered facilities of great importance, especially after the COVID-19 pandemic, and, therefore, choosing appropriate locations for new hospitals is very important. The research gap is motivated by the application of methods for predicting the suitability of a hospital location.

In this paper, a model that combined CFS, MLP, and GIS was proposed to optimize and rank 14 conditioning factors that affect hospital sites. The proposed model identified the factors that contribute to appropriate location selection and removed the inappropriate parameters observed to reduce the error in the model. The MLP model was chosen because it has recently been successfully applied in several fields of study and was established to be more promising than other ML models.

The MLP and AHP models were examined and compared in this paper to explore the effectiveness of some chosen factors that affect the ability to assess hospital site suitability. This study experimented with and validated the MLP model successfully. The MLP model was shown to function optimally and provided more consistent outcomes in terms of the validity of the study area. The outcome of this study showed that it is accurate to summarize that MLP is an acceptable model according to the validation values of CFS and AUC.

In conclusion, the performance of the MLP model showed that the proposed model is applicable and appropriate for assessing the suitability of a hospital site, and we believe that the MLP model should be used to assess the suitability of hospital sites, since it has been successfully used in different study area characteristics. Future studies may apply different ML models to an identical objective, combining advanced and ensemble techniques to enhance the capability of the models.

**Author Contributions:** Conceptualization, K.Y.A.; methodology K.Y.A.; software, K.Y.A.; validation, K.Y.A.; formal analysis, K.Y.A.; investigation. K.Y.A.; resources, K.Y.A.; data curation, K.Y.A.; writing—original draft preparation, K.Y.A.; writing—review and editing, K.Y.A., B.K., and N.U.; visualization, K.Y.A. and B.K.; supervision, A.R.M.S., A.F.A., and S.N.S.I.; funding acquisition, B.K. All authors have read and agreed to the published version of the manuscript.

**Funding:** This research received no external funding.

**Institutional Review Board Statement:** Not applicable.

**Informed Consent Statement:** Not applicable.

**Data Availability Statement:** The data used to support the findings of this study are available from the corresponding author upon request.

**Acknowledgments:** The authors would like to thank the Universiti Putra Malaysia and RIKEN AIP, Japan for providing facilities during this research.

**Conflicts of Interest:** The authors declare no conflict of interest.

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
