# Peer review of "Performance Evaluation of Hospital Site Suitability Using Multilayer Perceptron (MLP) and Analytical Hierarchy Process (AHP) Models in Malacca, Malaysia"

_sustainability, doi:10.3390/su14073731_

Round 1
Reviewer 1 Report
This paper presents the design of a location decision support system for potential hospitals, this study utilized multilayer perceptron (MLP) and analytical hierarchy process (AHP) models.
Before publishing this paper it is necessary to discuss some issues and make necessary corrections.
My concern is the redundancy of information in the layers related to flooding.
Decision-making information in this area comes simultaneously from many layers at the same time.
My doubts are raised by the reliability of the maps h i j k l all have identical legends.
The resulting raster distance from highway does not look reliable. What are the sharp discontinuities due to?
The manuscript contains numerous linguistic and editorial errors.
Author Response
Dear Reviewer,
Thank you for your useful comments and suggestions on the content and structure of our manuscript. We have addressed all comments (highlighted part in manuscript) and our response is attached.
Best Regards,
Dr. Kalantar

Reviewer 2 Report
Authors presented very important subject in their paper. Suitability analysis in recent time finds very important role in spatial planning and decision making. So, my recommendation is to elaborate it more according to different literature in introduction part. It is very important to mention (1977) Methods for Generating Land Suitability Maps: A Comparative Evaluation, Journal of the American Institute of Planners, 43:4, 386-400, DOI: 10.1080/01944367708977903
Also, authors should explain what represents most suitable and what less suitable in conditioning factors. E.g. distance from the road/highway - what distance is the most suitable? Vicinity of highway probably is not very attractive for the hospital due to the noises, dust etc. But it should be in appropriate distance for ER. Same for the slope, it is not clear what authors consider the most suitable for building hospitals. So, please elaborate such evaluation method for every conditioning factor, while taking into account social, geotechnical and engineering requirements.
Please check the adequacy of used color ramp, maps are not clear enough.
Distance from the river is misspelled (figure 3j).
Author Response

(The authors gave the same response as above.)

Reviewer 3 Report
The introduction should contain research questions and the research goal, as well as discuss the structure of the article.
Related Studies should be covered in a separate section (eg Literature review). A look-up table on the uses of MCDA methods in the hospital site selection problem should be added. The authors in this review refer to a very large number of articles, but they do it very sparingly. Instead of such short descriptions, it is better to refer to these articles in the form of a table with the MCDA methods used, decision problem, etc. A similar table can be presented for ML applications in the site selection problem.
In the description of the AHP method, the authors provide a non-standard course of the calculation procedure that uses column sums instead of right eigenvector. It should be mentioned in the article that this is a non-standard procedure and may produce different results than right eigenvector [1].
The article lacks a meaningful justification for the methodological choices made. Why was the AHP method used, and not, for example, ANP [2] or other MCDA methods? Why were CFS used and not, for example, FCBF [3], Symmetrical uncertainty [4] or Significance attribute [5] methods? Why was MLP used and not, for example, naive Bayes classifier [6] or random forest [7]?
Conclusions should indicate the directions of further research and research limitations.
[1] Ziemba, P., WÄ…tróbski, J., Jankowski, J., & Piwowarski, M. (2016). Research on the Properties of the AHP in the Environment of Inaccurate Expert Evaluations. Selected Issues in Experimental Economics, 227–243. doi:10.1007/978-3-319-28419-4_15
[2] Saaty, T.L.; Vargas, L.G. Decision Making with the Analytic Network Process: Economic, Political, Social and Technological Applications with Benefits, Opportunities, Costs and Risks; International Series in Operations Research & Management Science; 2nd ed.; Springer US, 2013.
[3] Yu, L.; Liu, H. Feature Selection for High-Dimensional Data: A Fast Correlation-Based Filter Solution.; January 1 2003; Vol. 2, pp. 856–863
[4] Yang, Q.; Shao, J.; Scholz, M.; Plant, C. Feature Selection Methods for Characterizing and Classifying Adaptive Sustainable Flood Retention Basins. Water Research 2011, 45, 993–1004, doi:10.1016/j.watres.2010.10.006
[5] Ahmad, A.; Dey, L. A Feature Selection Technique for Classificatory Analysis. Pattern Recognition Letters 2005, 26, 43–56, doi:10.1016/j.patrec.2004.08.015
[6] Ziemba, P., Becker, J., Becker, A., Radomska-Zalas, A., Pawluk, M., & Wierzba, D. (2021). Credit Decision Support Based on Real Set of Cash Loans Using Integrated Machine Learning Algorithms. Electronics, 10(17), 2099. doi:10.3390/electronics10172099
[7] Breiman, L. Bagging Predictors. Machine Learning 1996, 24, 123–140, doi:10.1023/A:1018054314350
Author Response

(The authors gave the same response as above.)

Round 2
Reviewer 1 Report
Paper is improved, anyway please format the values in legends on distance maps [h, i j]. I think 1 or 2 decimal places should be enough.
Author Response
Dear Reviewer,
Thank you for your comment. Please find the attached file.
Best Regards,
Dr. Kalantar

Reviewer 3 Report
The article has been corrected in line with the comments. Only one point remains:
- please renumber the sections: 2. Literature review, 3. Materials and Methods, etc.
Author Response

(The authors gave the same response as above.)
